# Evaluation of Quantitative Platforms for Single Target Mass Spectrometry Imaging

**DOI:** 10.3390/ph15101180

**Published:** 2022-09-23

**Authors:** Andrew P. Bowman, James Sawicki, Nari N. Talaty, Wayne R. Buck, Junhai Yang, David S. Wagner

**Affiliations:** AbbVie, Inc., 1 Waukegan Rd, North Chicago, IL 60064, USA

**Keywords:** mass spectrometry imaging, mass spectrometry, antibody–drug conjugate, small molecule drug, protein degrader, MALDI, DESI, QQQ, TimsTOF, FT-ICR

## Abstract

(1) Imaging of pharmaceutical compounds in tissue is an increasingly important subsection of Mass Spectrometry Imaging (MSI). Identifying proper target engagement requires MS platforms with high sensitivity and spatial resolution. Three prominent categories of drugs are small molecule drugs, antibody-drug conjugate payloads, and protein degraders. (2) We tested six common MSI platforms for their limit of detection (LoD) on a representative compound for each category: a Matrix-Assisted Laser Desorption/Ionization (MALDI) Fourier Transform Ion Cyclotron, a MALDI-2 Time-of-Flight (ToF), a MALDI-2 Trapped Ion Mobility Spectrometry ToF, a Desorption Electrospray Ionization Orbitrap, and 2 Atmospheric Pressure-MALDI Triple Quadrupoles. Samples were homogenized tissue mimetic models of rat liver spiked with known concentrations of analytes. (3) We found that the AP-MALDI-QQQ platform outperformed all 4 competing platforms by a minimum of 2- to 52-fold increase in LoD for representative compounds from each category of pharmaceutical. (4) AP-MALDI-QQQ platforms are effective, cost-efficient mass spectrometers for the identification of targeted analytes of interest.

## 1. Introduction

Mass spectrometry imaging (MSI) has pushed the boundary of pharmaceutical studies by allowing for ever more precise understanding of the distribution of drugs [1,2], metabolites [3,4], and endogenous products [5,6] within tissue sections. Advancements in MSI technology have improved both the sensitivity and resolution with sensitivity routinely approaching 1 microgram of analyte (for *m*/*z* ~300–800) per gram of tissue [7,8], and resolution reaching 5 μm on commercial platforms [9,10]. However, the conundrum of increasing resolution at the expense of sensitivity is still an issue [11,12,13]. With many cells being 10–20 μm in diameter, current technology can begin to distinguish between individual cells [9,10], in addition to different cell types within a given tissue [14,15,16], vis à vis distinguishing between inner medulla, inner stripe, glomeruli, and the interstitium of kidneys [17]. Further, subcellular MSI resolution becomes possible [18,19,20,21], allowing for the identification of specific compartments, such as the dense accumulation center of cholesterol in phagocytes involved in multiple sclerosis [17].

Often the focus in MSI studies is spread across multiple molecular species, attempting to piece together an overview of biology by aggregating all their distribution profiles [22,23]. This is often the case when searching for unknowns or casting a wide net for gaining the most information. However, within the field of drug metabolism and pharmacokinetics, the priority is, typically, on a single species of interest [24]. For targeted imaging studies, the intent is often to push both sensitivity and spatial resolution to their limits, to analyze the exact distribution of a compound within a tissue down to both the cellular and subcellular compartment [25,26]. Concomitantly, instrumentation that has the highest sensitivity and spatial resolution are prioritized, with the speed of acquisition also being a prominent factor. Often, the need for rapid turnaround in sample analysis is important [25], as well as the potential for degradation of analyte over time [22,24], or for the loss of matrix in matrix-assisted MSI techniques.

To push the envelope of targeted drug studies, the instrument would have single molecule detection, subcellular spatial resolution, fast acquisition speed, and broad molecular coverage in a single scan. Additionally, the perfect instrument could uniquely identify multiple species of interest or multiple ranges of potential *m*/*z* to accommodate both metabolites and small drugs, as well as peptides and potentially small proteins. No single mass analyzer can cover every molecule, nor can any ion source. However, there is still utility in finding a platform of mass analyzer and ion source that allows for broad applicability and utility to Drug Metabolism and PharmacoKinetics (DMPK). In such an environment, we can often ignore broad molecular coverage in favor of single-target analysis, relaxing some of the constraints on mass spectrometer choice.

There are 3 broad categories of MSI ionization sources that see the most use: secondary ion mass spectrometry (SIMS) [16,19], matrix-assisted laser desorption ionization (MALDI) [9,21,27,28,29,30], and desorption electrospray ionization (DESI) and its emergent developments [1,6,10,14,31]. SIMS is the highest spatial resolution technique available, achieving nanometer-scale resolution, with such resolution accompanied by high rates of fragmentation [32,33]. Lower resolution (only by contrast to other SIMS modalities) options exist, though the upper *m*/*z* limit for SIMS is still lower than for other methodologies [34,35,36]. By contrast, MALDI has achieved commercially available 5 micrometer resolution, with research-instruments approaching sub-micron resolution [30,37]. MALDI sources come in two main subtypes by the pressure of source: vacuum and atmospheric-pressure (AP). Vacuum-based MALDI instruments are more common than AP-MALDI sources and are commercially available as complete systems with time-of-flight (TOF) and Fourier Transform mass analyzers. AP-MALDI sources are becoming increasingly common due to the gentler ionization process vs. vacuum-MALDI [38,39], as well as the fact that the volatile matrix compounds needed for MALDI are less likely sublimed at atmospheric pressure, allowing for longer experiments [40]. Commercially available MALDI-2 systems are also now available for vacuum-MALDI instruments. MALDI-2 shows significant enhancements in ionization efficiency of vacuum-MALDI platforms by adding a second “post-ionization” laser parallel to the tissue surface [17,21,41]. DESI and DESI-like sources are advantageous due to their ability to generate multiply charged precursors, greatly enhancing the size of molecules available for analysis. While DESI itself is limited to approximately 50 μm in spatial resolution, nano-DESI and other successor technologies have approached the level of commercial MALDI sources at 10 µm [31,42].

Mass analyzers cover a wide terrain of mass resolution and analysis speed, as well as offering potential confirmatory techniques, *a la* tandem methods (MS/MS, MS^2^, MS^n^). Triple quadrupole (QQQ) instruments can uniquely identify species via MS^2^ experiments, with effectively no background noise through multiple-reaction monitoring (MRM) acquisition mode [43,44]. In MRM-mode, they are also fast instruments, with scan times on the order of milliseconds, slowed only by the time of reaction in the collision cell (typically 50 ms or less). However, most QQQ platforms are limited to *m*/*z* of 4000 or less, significantly below the *m*/*z* of proteins. This is not a limitation for DESI sources, but the primarily singly charged MALDI spectra would be unable to see such high mass ions. Isolation of species by MS^2^ can additionally be hampered by three constraints: a species of interest must be fragmentable, the fragment must retain the charge, and the fragment needs to be unique either alone or in tandem with other monitored fragments.

Specificity of identification can also come through the addition of an ion mobility stage between the ion source and the mass analyzer. Drift tube [45], travelling wave [46], trapped ion mobility spectrometry (TIMS) [47] cells are all compatible with imaging sources such as MALDI or DESI. While the exact details vary based on the ion mobility cell used, functionally all separate gas-phase ions by their collisional cross section.

As an alternative to MRM studies focused on a single (or multiple) transitions of a single target of interest, high resolution mass spectrometers can potentially identify analytes of interest by accurate mass, retaining other information that can be useful in identifying cell type as well as multiple precursor and metabolite masses simultaneously. Time-of-Flight instruments readily achieve mass resolutions of 60,000, with 200,000 having recently been reported [10]. These are also the fastest mass analyzers available, performing full scans in microseconds, allowing them to match well with ion mobility technologies [48,49,50]. Fourier transform instruments have even higher mass resolutions, with Orbitrap instruments reaching 1 million [51] and ion cyclotron resonance instruments achieving resolutions exceeding 2.7 million [52]. However, FT instruments are significantly slower than either QQQ or TOF instruments, which is problematic both for throughput and for sustainability when paired with vacuum-MALDI ion sources.

The final challenge to find an MSI platform for a DMPK environment lies in quantitation and determining the limit of detection (LoD) across multiple platforms. Traditionally, the limit of detection has been determined by droplet deposition onto a thin tissue section. This is quick and simple as a method but is prone to multiple types of error: the extraction efficiency of the surface-spotted analyte is significantly higher than the analytes within the tissue, and the “coffee ring” effect of dried droplets make absolute quantitation difficult [27,28]. More recently, multiple groups have reported the use of spiked tissue homogenate, either as a plug of material or separated into columns [53,54,55]. This has the advantage that the analyte should be evenly distributed throughout the homogenate block, and that the environment for the analyte should be the same as all the endogenous products within the tissue. There are some concerns about the change in microenvironment created through the rupturing of the cells in the homogenization process, but experimental evidence shows that the response curves generated through spiked homogenates more closely resemble the response from liquid-chromatography-based methods. An alternative has been offered in the tissue extinction coefficient, which offers similar quantitation on tissue [56].

Herein we present an analysis of four mass spectrometry platforms to determine the best platform for a DMPK environment: A vacuum-MALDI, FTICR platform; a vacuum-MALDI-2 platform, with and without TIMS; an AP-MALDI, QQQ platform; and a DESI-Orbitrap platform. While speed of analysis is a factor, the most important parameters are the limit of detection and the spatial resolution for a single target of interest. Tissue mimetic models of 3 representative compounds, felodipine (a small drug), monomethyl auristatin E (MMAE, an antibody-drug conjugate payload), and VZ-185 (a protein degrader), were prepared and analyzed using the above platforms using 3 matrices, 2,5-dihydroxybenzoic acid (DHB), 2,6-dihydroxyacetophenone (DHA), and FleXmatriX (FleX).

## 2. Results

Full dataset comparisons are available in Appendix A. Ion images from each platform have been added to the supplement, demonstrating the relative homogeneity of the response from each tissue homogenate (Appendix A). Calculations of the standard deviation of the LoB and LoD for felodipine with DHB matrix on the TimsTOF MALDI-2 are available in the Appendix A (Appendix A). For 5 samples spread across 1.5 months (23 March to 2 May), RSD of LoD varied from 20.60 to 31.70 percent.

For the small drug felodipine, the improvement of the AP-MALDI-QQQ-MRM platform is 2.54-fold enhancement of the LoD over the FT-ICR, and 2.61 over the MALDI-2 TimsTOF (Table 1) on the 6500 platform. The 6500+ is 2.09-fold better than the 6500. DHB was the matrix that provided the lowest LoD for all tested conditions except for magnitude mode (the default operational mode) FT-ICR, where it was overtaken by FleX matrix. FleX was the only matrix tested on the 6500+ due to time constraints. While DHB performed better than FleX matrix on the 6500 platform, FleX on the 6500+ platform outperformed DHB on the 6500.

MMAE, an antibody-drug-conjugate payload, offered the lowest gain for the AP-MALDI-6500-MRM platform over the competing platforms, but still improved the LoD by 1.80-fold vs. the next-best platform, the FT-ICR in magnitude mode (Table 2). By contrast, the AP-MALDI-6500+-MRM platform was 7.59-fold more sensitive than the 6500-MRM platform, with the lowest LoD of all compounds tested in this study at 3.74 µg/g.

VZ-185, a commercially available protein degrader, shows the greatest improvement of AP-MALDI-QQQ over our other imaging platforms, with a 35-fold increase over CASI on the FT-ICR, 52-fold over FT-ICR absorption mode, and 111-fold over the TimsTOF with TIMS separation (Table 3). Concomitantly, the increase from 6500 to 6500+ was the smallest improvement at 1.79-fold. Unlike for either MMAE or felodipine, the use of the TIMS separation significantly improves the LoD of the TimsTOF (1.94-fold). Absorption mode for VZ-185 also shows significant improvements over magnitude mode, improving detection from DHB matrix by almost 3-fold. Again, our DESI platform failed to perform in competition with any tested MALDI platform.

Representative images of the mimetic models have been included in the Appendix A (Appendix A). The images are roughly homogeneous, though some cracking is evident from the thaw mounting process.

## 3. Discussion

An important note across all 3 analytes is that the choice of matrix is of great importance. Even though all 3 of these matrices are based on a dihydroxybenzene ring, their response for the molecule of interest varies significantly. LoD varied in-platform by up to ~10-fold between DHB, DHA, and FleX matrices, and creates variation between platforms such that which platform has the lower LoD can change based on matrix chosen.

To potentially improve the LoD on the TimsTOF platform, we attempted isolation in both a CASI-like (broadband) and product ion scan operation modes. In our experimental setup, CASI-like isolation did not improve the LoD, and fragmentation significantly worsened the LoD. CASI-mode on the FT-ICR in magnitude mode decreased the LoD for felodipine, had no effect for MMAE, and improved the LoD for VZ-185. Use of the TIMS cell is also inconsistent. Across the 3 analytes, felodipine LoD was worsened by 1.86-fold with the use of TIMS, MMAE was not significantly affected, and VZ-185 was improved by 1.94-fold. Narrow-band scanning of the TIMS cell might help to improve utility in all 3 cases but carries the side-effect that the *m*/*z* range must correspondingly decrease. In targeted drug studies such as this, that can be of high utility, although, as for the improvements possible through absorption-mode in the FT-ICR, the increase is still lacking in comparison to the QQQ-platforms.

Absorption mode [57] followed a similar pattern to TIMS use on the TimsTOF platform. The LoD of felodipine was worsened using absorption mode, MMAE was not improved, and VZ-185 was significantly improved. Further experimentation with narrow isolation widths to improve phasing in CASI experiments should improve the response through that mode, although it is unlikely that it would approach the sensitivity of the QQQ-based platforms.

Interestingly, the TimsTOF and FT-ICR platforms both observed the same trend in improvement/degradation, where attempted isolation degraded signal intensity and LoD for the low mass felodipine, had no effect for MMAE, and improved significantly VZ-185. We tentatively posit that this is due to the strength of the isolation RF voltage and could likely be improved by systematic variation of the RF, though we conclude it is unlikely to improve either platform to the sensitivity of the QQQ-based platforms.

The DESI platform available to us in these experiments was overshadowed by all MALDI platforms, save the absorption mode FT-ICR data, owing to significant signal instability in that experiment. Research using a more modern nano-DESI platform will likely lead to significantly different results.

## 4. Materials and Methods

### 4.1. Materials

HPLC-grade Water, methanol, ethanol, chloroform, DMSO, and acetonitrile were acquired and used without further purification (Sigma-Aldrich, Burlington, MA, USA). 2,6-Dihydroxyacetophenone (MS-Grade, Sigma-Aldrich) and FleXMatrix (Bruker Daltoniks, Bremen, Germany) were purchased and used without further purification. 2,5-Dihydroxybenzoic Acid (99%, Sigma-Aldrich) was recrystallized twice with water. Standard ITO slides (Delta Technologies, Loveland, CO, USA) were cleaned with hexane and ethanol (sonication, 6 min) sequentially. Felodipine, monomethyl auristatin E (MMAE), and protein degrader VZ-185 (Sigma-Aldrich) were used without any further purification.

### 4.2. Biological Samples

Naïve rat liver was treated according to the following protocol. 1 g of tissue was rough chopped and placed in a 2.5 mL vial with ~5 stainless steel homogenizer beads (Biospec Products, Bartlesville, OK, USA). Vials were frozen at −80 °C for 90 s, then homogenized on a Mini-Beadbeater 24 (Biospec Products) for 90 s. Samples were frozen and homogenized for 10 cycles. 95 μL of liver homogenate was spiked with 5 μL of solution containing analyte standards with 20, 6.32, 2, 0.632, 0.2, and 0 mg/mL, resulting in concentrations of 1000, 316, 100, 31.6, 10, and 0 μg/g analyte/tissue. Spiked samples were then shaken on the homogenizer for 30 s to ensure uniform distribution. Analytes were dissolved in solvents chosen to aid complete dissolution: felodipine:EtOH, MMAE:EtOH, and VZ-185:DMSO.

### 4.3. Sample Preparation

Homogenate samples were pipetted into a prepared gelatin (15% *w*/*v*) mold with 6 individual pillars. 5 of the 6 pillars are square, with the final pillar being round to maintain orientation of thin tissue sections (Figure 1). Prepared homogenate blocks were then frozen at −80 °C until cutting. Samples were sectioned at 12 μm thickness with 4 sections mounted per ITO slide (Figure 2). Slides were then coated with one of 3 matrices, using the same standard parameters on a TM Sprayer (HTX Technologies, Chapel Hill, NC): 20 mg/mL in CHCl3:MeOH (2:1 *v*/*v*); flow rate, 0.12 mL/min; N2 pressure, 10 psi; spray-head temperature, 80 °C; spray-head velocity, 1200 mm/min; track spacing, 3 mm; number of layers, 8; drying time between layers, 30 s.

### 4.4. Instrumentation

MSI experiments were carried out on 4 different platforms: the TimsTOF FleX MALDI-2 qTOF (Bruker Daltoniks), the SolariX 2xR 7T FT-ICR (Bruker Daltoniks), a Q Exactive Plus Orbitrap (ThermoFisher Scientific, San Francisco, CA) with a DESI ionization source (Prosolia, Indianapolis, IN, USA), and 2 Triple Quad instruments, a 6500 and 6500+ (AB Sciex, Framingham, MA, USA) with an AP-MALDI UHR source (MassTech, Inc., Columbia, MA, USA).

Instrumental parameters for the TimsTOF platform were: MALDI-2 enabled, 20 laser shots per pixel, pixel size 20 μm, positive ionization mode. Full imaging datasets (200–1200 *m*/*z*) were acquired both with the TIMS module off and on, when on, the TIMS was operated from 0.8–1.96 1/k_0_ and 100 ms ramp time. Approximate speed of acquisition was ~25 pixels/s without the TIMS module, and ~9 pixels/s with the TIMS module.

SolariX full imaging datasets (200–1200 *m*/*z*) were collected with 50 laser shots per pixel, pixel size 50 μm, positive ionization mode, data size set to 512 k. This resulted in an acquisition rate of ~3 pixels/s. Additional datasets were acquired using Continuous Accumulation of Selected Ions (CASI) mode. Isolation windows were set at ±25 *m*/*z*, centered on the theoretical sodiated peak. Absorption mode spectra were calibrated immediately prior to acquisition using coffee extract, either in full MS mode from *m*/*z* 150–1500 at data size 512 k or as described for CASI mode above.

AP-MALDI QQQ datasets were collected with the source operating in continuous motion mode with the laser set to 5 kHz, pixel size of 20 μm. The QQQ platform was operated in Multiple Reaction Monitoring (MRM) mode with 3 transition states: 2 for the analyte in question and 1 for the matrix as a stability monitor over the length of the experiment. Collisional energy was optimized per analyte by dried droplet deposition on a clean ITO slide. MRM reactions were set for 50 ms with 5 ms between transitions, resulting in a pixel rate of ~6.06 pixels/sec. 6500 datasets were collected for all analytes and matrices, while 6500+ datasets were only collected for FleX matrix.

DESI-Orbitrap datasets were collected with the source operating in continuous motion mode at 1000 μm/s with the Orbitrap set to a resolution of 60,000, resulting in a pixel time of 200 ms and pixel size of 200 μm. Pixel acquisition rate is 5 pixels/s.

To test the robustness of the system, multiple runs of felodipine coated with DHB were tested on the TimsTOF MALDI-2 platform. 5 samples of DHB-coated tissue mimetic were analyzed across 1.5 months. Samples were analyzed at approximately 5000, 10,000, and 20,000 pixels per analyte concentration.

### 4.5. Data Treatment

Data from the TimsTOF, SolariX, and Orbitrap were loaded into SCiLS Lab (Bruker Daltoniks). Samples were normalized to the intensity of a known matrix cluster peak (*m*/*z* 380), with 99% hotspot removal. Samples were subjected to bisecting k-means, so that all further data processing is based solely on pixels that contain homogenate and not gelatin background. Average signal intensity and standard deviation of pixels were exported from SCiLS for final processing. Limit of Blank (*LoB*) and Limit of Detection (*LoD*) based on the following formulas:LoB=Avg(0)+1.96×StdDev(0)LoD=LoB+1.96×StdDev(low)
where *Avg*(0) is the average signal intensity of the *m*/*z* of the analyte in the blank, *StdDev*(0) is the standard deviation of the signal intensity of the analyte *m*/*z* in the blank, and *StdDev*(*low*) is the standard deviation of the analyte in the lowest concentration (i.e., 10 μg/g). 1.96 × *StdDev* is the 95% confidence interval. *LoD* and *LoB* were then converted to concentrations via simple linear regression. Signal intensities and standard deviations from SolariX and DESI-QE platforms were converted from 50 and 200 μm pixels, respectively, to 20 μm. This was achieved as a first-order comparison by decreasing the signal intensity by the area ratio (vis à vis 0.16 and 0.01) and decreasing the standard deviation by the square root of the area ratio (0.4 and 0.1).

Samples from the AP-MALDI QQQ platform were treated in a similar method to all other datasets, where homogenate containing pixels were defined as all pixels that had signal intensities greater than 0. Samples also ignored all pixels >99th percentile. Otherwise, samples were treated the same as all other platforms.

## 5. Conclusions

We found in these experiments that the use of an AP-MALDI-QQQ-MRM platform showed significantly better LoD for 3 major drug categories (small drug, antibody-drug conjugate, and protein degrader). Further, the choice of which matrix to use can offer significant improvements of 10-fold lower limit of detections. The field is ready for a greater, in-depth study on the applicability of different matrices to different compounds to improve LoDs. Additional utility can also be found in the use of sublimation rather than wet spray, allowing for much smaller, more uniform crystal sizes to probe at the limit of our spatial resolution, rather than the 20 µm chosen here. However, we cautiously promote the utility of the relatively inexpensive AP-MALDI-QQQ platform vs. either TimsTOF-FleX, SolariX, or DESI-Q Exactive platforms, finding that they are worthy investments for laboratories whose goal is analysis at the limit of resolution and sensitivity.

## Figures and Tables

**Figure 1 pharmaceuticals-15-01180-f001:**
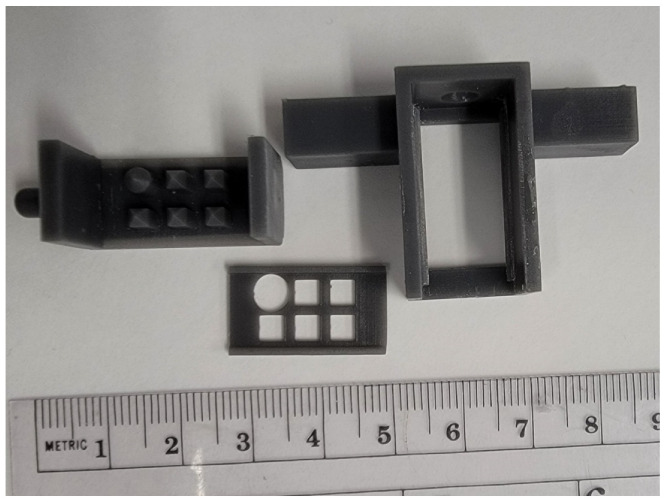
Plastic mold for gelatin mold. Lifter plate (**center**) is inserted into pillar mold (**left**), which is slotted into outer mold (**right**).

**Figure 2 pharmaceuticals-15-01180-f002:**
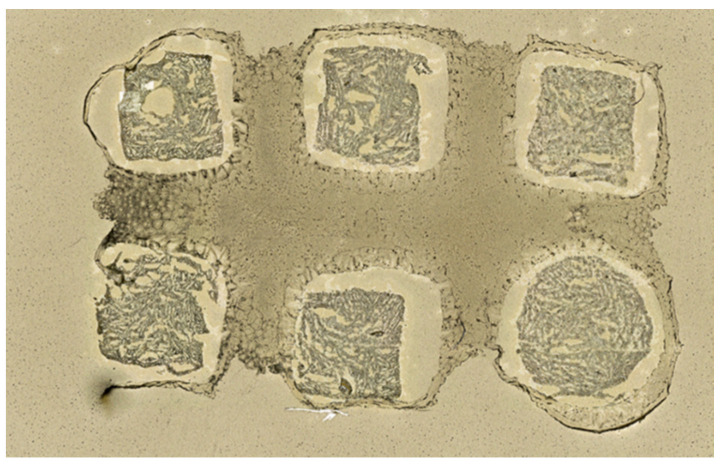
Representative section of tissue mimetic model. Samples are relatively homogenous, though the difference in shrinkage between the homogenate and gelatin causes it to separate after transfer to the slide.

**Table 1 pharmaceuticals-15-01180-t001:** Comparison of lowest limit of detection across the imaging platforms, adjusted to 20 µm^2^ pixels. While 3 matrices were used in the initial experiment, DHA did not provide the lowest LoD for felodipine. R^2^ values of the linear regression are also included, with most R^2^ > 0.99, but none lower than 0.96.

Platform	Matrix	R^2^	Limit of Detection (µg/g Felodipine/Liver)
AP-MALDI 6500+	FleX	0.9946	53.5
AP-MALDI 6500	DHB	0.9757	111.83
FT-ICR Magnitude	FleX	0.9943	283.75
MALDI-2	DHB	0.9926	292.29
MALDI-2 w/TIMS	DHB	0.9960	542.76
FT-ICR CASI	DHB	0.9990	647.39
FT-ICR Absorption CASI	DHB	0.9820	982.17
DESI Orbitrap	N/A	0.9948	1968.17
FT-ICR Absorption	DHB	0.9630	3108.88

**Table 2 pharmaceuticals-15-01180-t002:** Comparison of lowest limit of detection across the imaging platforms, adjusted to 20 µm^2^ pixels. While 3 matrices were used in the initial experiment, DHA did not provide the lowest LoD for MMAE. R^2^ values of the linear regression are generally poorer than for felodipine, but all are above 0.9.

Platform	Matrix	R^2^	Limit of Detection (µg/g MMAE/Liver)
AP-MALDI 6500+	FleX	0.9953	3.74
AP-MALDI	FleX	0.9812	28.38
FT-ICR Magnitude	FleX	0.9939	51.15
FT-ICR CASI	DHB	0.9993	57.64
FT-ICR Absorption CASI	DHB	0.9968	59.66
MALDI-2	FleX	0.9923	65.40
MALDI-2 w/TIMS	DHB	0.9650	79.89
FT-ICR Absorption	DHB	0.9098	80.16
DESI Orbitrap	N/A	0.9711	998.28

**Table 3 pharmaceuticals-15-01180-t003:** Comparison of lowest limit of detection across the imaging platforms, adjusted to 20 µm^2^ pixels. While 3 matrices were used in the initial experiment, FleX did not provide the lowest LoD for VZ-185. Except for the CASI and absorption mode FT spectra, all reported lowest LoD with DHA. Despite this, the AP-MALDI-6500+ platform exceeded all other platforms by ~2-fold while only being run with FleX matrix. R^2^ values of the linear regression are generally poorer than for felodipine, but all are above 0.92.

Platform	Matrix	R^2^	Limit of Detection (µg/g VZ-185/Liver)
AP-MALDI 6500+	FleX	0.9950	7.69
AP-MALDI	DHA	0.9398	13.76
FT-ICR CASI	DHB	0.9997	475.77
FT-ICR Absorption	DHB	0.9858	715.13
FT-ICR Absorption CASI	DHB	0.9864	798.05
FT-ICR Magnitude	DHA	0.9417	1370.84
MALDI-2 w/TIMS	DHA	0.9655	1534.08
MALDI-2	DHA	0.9905	2975.61
DESI Orbitrap	N/A	0.9264	7126.02

## Data Availability

Data is contained within the article and Appendix A.

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
