# Peer review of "Evaluation of Quantitative Platforms for Single Target Mass Spectrometry Imaging"

_pharmaceuticals, 2022, doi:10.3390/ph15101180_

Round 1

Reviewer 1 Report

The subject of the manuscript is interesting and innovative. The authors tested six MSI platforms for the detection limit of three representative drugs: felodipine (a small drug), monomethyl auristatin E (MMAE, an antibody-drug conjugate payload), and VZ-185 (a protein degrader). The manuscript is generally well written. The aim of the work and the research methodology are precisely described. The conclusions are correct. The authors presented the limits of detection of the above-mentioned drugs on the tested MSI platforms. However, the authors do not write anything about the precision and repeatability of the obtained LOD values. Therefore, authors must supplement the manuscript with these informations.

Reviewer 2 Report

Bowman et al. present work evaluating different mass spectrometry configurations for analyzing different size molecules by mass spectrometry imaging.  The manuscript is well written and succinct.  The authors suggest using a configuration of AP-MALDI-QQQ for superior limits of detection and demonstrate the importance of selecting a proper matrix. I only have minor comments and suggestions.

Lines 9-10: Spell out three and six.

Lines16-17: “…by a 2- to 52-fold increase in LoD for…”

Add a space before [references].

Line 30: Check spacing before “with.”

Line 34:  Remove “the” before subcellular.

Line 100: LoD to be consistent with rest of manuscript.

Line 117: What is a MALDI-2 platform? Add a sentence in the intro. Also, for non-experts, what is TIMS? The acronym is not defined nor is it explained.

Table 1: Why was FleX used as the matrix for the 6500+?

Line 153-154: It is said that DHA provides the lowest LoD across all platforms, but Table 3 for FT-ICR configurations, the lowest is DHB.

Table 3 does not a have a legend.

Line 158: “…choice of matrix is of…”

Line 258: No new paragraph.

Reviewer 3 Report

It would be necessary to review and broaden the bibliographical references since many citations relevant to the technique are missing, such as (merely illustrative and not exhaustive):

https://www.annualreviews.org/doi/10.1146/annurev-physchem-061020-053416

https://pubs.acs.org/doi/10.1021/acs.analchem.7b04733

https://pubs.acs.org/doi/10.1021/ac504543v

Likewise, it would be essential to present a processed image where the performance of the different techniques is visualized.

Round 2

Reviewer 1 Report

The authors revised the manuscript in line with the comments in the review. Therefore, the manuscript can be published as is.